# Drivers of Foliar Fungal Endophytic Communities of Kudzu (*Pueraria montana* var. *lobata*) in the Southeast United States

**Maryam Shahrtash** [1,2] **and Shawn P. Brown** [1,2,*]

[1] Department of Biological Sciences, The University of Memphis, Memphis, TN 38152, USA; mshhrtsh@memphis.edu
[2] Center for Biodiversity Research, The University of Memphis, Memphis, TN 38152, USA
[*] Correspondence: spbrown2@memphis.edu; Tel.: +1-901-678-3596

**Abstract:** Fungal endophytes play important roles in plant fitness and plant–microbe interactions. Kudzu (*Pueraria montana* var. *lobata*) is a dominant, abundant, and highly aggressive invasive plant in the Southeast United States. Kudzu serves as a pathogen reservoir that impacts economically important leguminous crops. We conducted the first investigations on kudzu fungal endophytes (Illumina MiSeq—ITS2) to elucidate drivers of endophytic communities across the heart of the invasive range in the Southeast United States (TN, MS, AL, GA). We tested the impacts of multiple environmental parameters (Chlorophyll, $NO_3^-$, $K^+$, soil pH, leaf area, host genotype, traffic intensity, and geographic location) on foliar endophyte communities. Endophytic communities were diverse and structured by many factors in our PerMANOVA analyses, but location, genotype, and traffic (proxy for pollution) were the strongest drivers of community composition ($R^2 = 0.152$, $p < 0.001$, $R^2 = 0.129$, $p < 0.001$, and $R^2 = 0.126$, $p < 0.001$, respectively). Further, we examined the putative ecological interactions between endophytic fungi and plant pathogens. We identify numerous OTUs that are positively and strongly associated with pathogen occurrence, largely within the families Montagnulaceae and Tremellales incertae sedis. Taken together, these data suggest location, host genetics and local pollution play instrumental roles in structuring communities, and integrative plant management must consider these factors when developing management strategies.

**Keywords:** Illumina MiSeq; fungal communities; pathogen facilitation; kudzu; invasive plants; endophytes; mycobiome

## 1. Introduction

Invasive plant species pose serious threats to landscapes and native ecosystems in the United States. Invasive plants are estimated to have a footprint of ca. 100 million ha. in the United States [1], and the average rate of spread is estimated as high as 14% per year [2]. Invasive plant management expenditures in the United States have been estimated at over 120 billion USD annually [3]. Often, invasive plants exist in isolation from co-evolved pests and pathogens that can suppress populations within their native ranges. Consequently, there can be very few impediments of successful colonization and expansion in invaded territories [4]. Further, phenotypic plasticity [5] and increased allelopathic potential [6] are often factors that allow plants to outcompete natives in their introduced range. Further, invasive plants can alter environmental conditions [7], suppress native species [8], and facilitate pathogen spillover events onto major crop species [9]; thus, understanding invasive plant mycobiomes is a crucial first step in developing integrative mitigation strategies.

Invasive plants can inhibit native species and facilitate the continuing spread of invasive plants through allelopathy [10], whereby invasive plants alter their environment by releasing various

allelochemicals including alkaloids, flavonoids and cyanogenic compounds, which results in the liberation of soil nutrients and water resources that they can preferentially utilize whilst inhibiting nearby native plant growth [11,12]. For example, ragweed (*Ambrosia artemisiifolia*) produces allelochemicals that have inhibitory effects on the seed germination and seedling growth of nearby plants [13]. Further, changes in soil microbial communities following the establishment of invasive species can elicit higher nitrification rates, which can shift competitive outcomes in favor of invasive plants against native plants [14].

The probability of invasive plant establishment and the rate of species expansion is often a function of geographic and ecological factors, which can determine the severity of invasion [15]. Ecological factors include light, temperature, water, wind, and atmospheric conditions and can play important roles in the distribution of invasive species and their success of establishment, spread, and persistence [16]. Additionally, ecosystem processes, including soil hydrology and nutrient cycling, can be altered by invasive plants [17–20]. These modifications of community structure and ecosystem functions can jeopardize the ecosystem stability and facilitate subsequent invasions [20,21].

Some invasive plants can serve as a reservoir or alternative host for pathogens and pests [22], which may facilitate disease persistence and spillover events into native systems. This significantly influences disease incidences on major crops [23]. For example, *Berberis vulgaris*, a widespread invasive barberry species, may act as a potential alternate host of the wheat stem rust pathogen, *Puccinia graminis* f. sp. *tritici*, in the Pacific Northwest of the US as long as germination conditions are met [24]. The highly invasive plant kudzu (*Pueraria. montana* var. lobata [Willd.] Maesen & S. Almeida) (Fabaceae) is a major pathogen reservoir for leguminous crops in the US which causes multibillion-dollar losses for producers annually. It has been suggested that kudzu may have been instrumental in the rapid spread of the devastating invasive Asian soybean rust (*Phakospora pachyrhizi*) across the US [25], which was first observed on kudzu in the United States in 2005 [26]. Further, using landscape-scale pathogen spillover models, it has been suggested that even small infected patches of kudzu are sufficient to maintain active epidemic levels of soybean rust [27].

Kudzu is a perennial, semi-woody, leguminous vine with a growth rate of up to 18 m in a single growing season [28]. To slow down soil erosion and maintain soil nutrients, the government encouraged farmers to extensively plant kudzu as a forage crop and soil stabilizer during the early 20th century across the southeastern part of the United States [29,30]. It was extensively planted on ca. one-million hectares between 1920 and 1950 [31]. The current introduced a range of kudzu ranges from New York to Florida and westward to central Oklahoma and Texas, with recent occurrences in Washington and Oregon; the heaviest infestations can be found in Alabama, Georgia, and Mississippi [32,33].

By 1953, the US Department of Agriculture recognized that kudzu was spreading too rapidly and removed it from its list of recommended cover plants [34] and labeled it federally as a noxious weed in 1997 [35]. Approximately 810,000 ha of the Southeast United States is challenged with kudzu [33], and, in the forestry industry, kudzu is responsible for loses between $100 to $500 million per year of productive land [30]. Additionally, costs to electric companies to control kudzu on utility poles have been estimated at $1.5 million per year [36]. It has been reported that kudzu populations in the United States show high genetic diversity and low genetic differentiation due to gene exchange and recombination [37] and high clonal reproduction [38]. Kudzu can outcompete native species and prevent their growth and survival, thus altering community dynamics [39,40]. In some parts of the Southeast US, kudzu has been implicated as a major contributing factor that has driven some native plants, such as *Trillium reliquum*, to be listed as endangered or threatened [40]. Further, kudzu strongly contributes to atmospheric pollution, as it has been demonstrated to emit large amounts of both nitric oxide and isoprene, which strongly contribute to ozone production [41].

Kudzu can be eradicated with persistent (multi-year) treatment with herbicides, livestock grazing, prescribed burning, and disk harrowing [33]; however, control through chemical means can take 10 years or more of repetitive herbicide applications [42] which may be prohibitively costly for some land managers. Biological control of kudzu has recently been demonstrated as a possible management

strategy but only with concurrent herbicidal treatments [43]. Boyette et al. [42] demonstrated that the generalist fungal pathogen *Myrothecium verrucaria* (Albertini and Schwein; Family Stachybotryaceae), which attacks leaves and stems and has as high as a 95% to 100% infection rate within 14 days of inoculation, can be used in conjunction with herbicides to suppress kudzu. However, this combination does not always elicit kudzu mortality, as seemingly dead kudzu can recover in a few years [44]. While bioherbicidal applications have been advocated for, implementation of these bioherbicides and integrated control programs have been slow [44].

Plants harbor diverse microbial communities that colonize their internal compartments without causing any harmful effect on host plants [45]. Fungal endophytes are often defined as taxa that live within plant tissue and are asymptomatic on their hosts [46], thus differentiated from surface-adhered epiphytes. This encompasses fungi that exhibit associations with plants that are neutral and commensal or may be dormant saprobes or latent pathogens that are not yet presenting symptoms [45]. Endophyte colonization can occur within or between plant generations; vertically through seeds [47] or pollen grains; and horizontally via soil- or air-borne spores [48]. These endophytes can reside within cells, in intercellular spaces, or in the vascular system [49]. Diverse tissue wounds occurring as a result of plant growth, stomatal openings [50], lenticels [51], and germinating radicles [52] are other common sites through which endophytes can enter plants.

Plant endophytic communities are controlled by many factors, including plant genotype and growth stage [53], growing season [54], developmental stage [55], plant host and geographic location [56], soil type [57], host plant nutrient status [58], host fertilization strategies [59], and agricultural practices [60]. Mutualisms can also play crucial roles in some invasion processes [61]. There is evidence that invasive plant interactions with mutualists can promote invasive growth over native plants [62,63]. This can be particularly problematic when examining invasive microbe interactions of plants that are closely related to native crop production plants. The close relationship between microbial community composition and plant genetic relatedness is likely due to greater similarity in physiological and biochemical characteristics among closely related species, such as root exudate chemistry [64]. Through the plant–soil feedback hypothesis [65], invasive plants can change soil microbial communities such that they can facilitate the plants' invasion success [66]. This can then mediate shifts in soil chemistry and properties which further favors plant establishment [15].

Plant endophytes can inhibit (pathogen antagonism) or promote (pathogen facilitation) plant diseases when susceptible plants, virulent pathogens, and favorable environmental conditions are present [67,68]. For example, foliar fungal endophytes of *Fallopia japonica* (Japanese knotweed) have been demonstrated to have faciliatory effects on *Phomopsis* spp. and have antagonist effects on *Alternaria* spp. and *Phoma* spp. [69]. Endophyte modification of plant disease development is dependent on abiotic and biotic factors as well as host and pathogen genetics, however, a full mechanistic understanding of these actions is mostly not achieved. Numerous studies indicate that interactions between endophytes and plants can help protect plants against several biotic and abiotic stresses, via hyperparasitism, competition or antibiosis [70,71]. However, some endophytes might also produce metabolites that can promote pathogen development [67]. Understanding the dynamics of the phyllosphere mycobiome may allow us to modify these communities to favor successful colonization and growth of taxa that facilitate pathogen virulence, which may be a novel method of integrated pest management of invasive plants. Exploring foliar endophytic communities in invasive plants to identify the drivers of community assembly will further our understanding of plant–microbe interactions within invasive plants. Identifying fungal endophytes that are potentially faciliatory of pathogens (that co-occur) is a promising approach that can lead to microbiome manipulations to actively manage kudzu and other invasive plants. The main goals of this study were to (1) elucidate the major drivers of foliar fungal endophytic assembly in kudzu and (2) investigate potential facilitatory or inhibitory associations between true endophytes and putative pathogens.

## 2. Materials and Methods

*Sample collection*—Leaves were collected from 38 sampling sites (individual plants) across the heart of the historic introduced range of kudzu in the US across the states of Tennessee, Mississippi, Alabama, Georgia (3–7 July 2018; Figure 1, Table S1). Locations were selected opportunistically by observing kudzu plants on roadsides and where accessible (to prevent trespassing on private land) leaves were sampled. At each site, three fully formed and visually disease-asymptomatic leaves from each plant were collected (114 total leaves). With our early July sampling, kudzu leaves were young and necrotrophic pathogen incidence remained low at our sampling sites, but leaf spot (e.g., *Septoria* spp.) was noticeable. Collected leaves were placed into clean, individually labeled zip-top plastic bags placed on ice until processed (within 12 h).

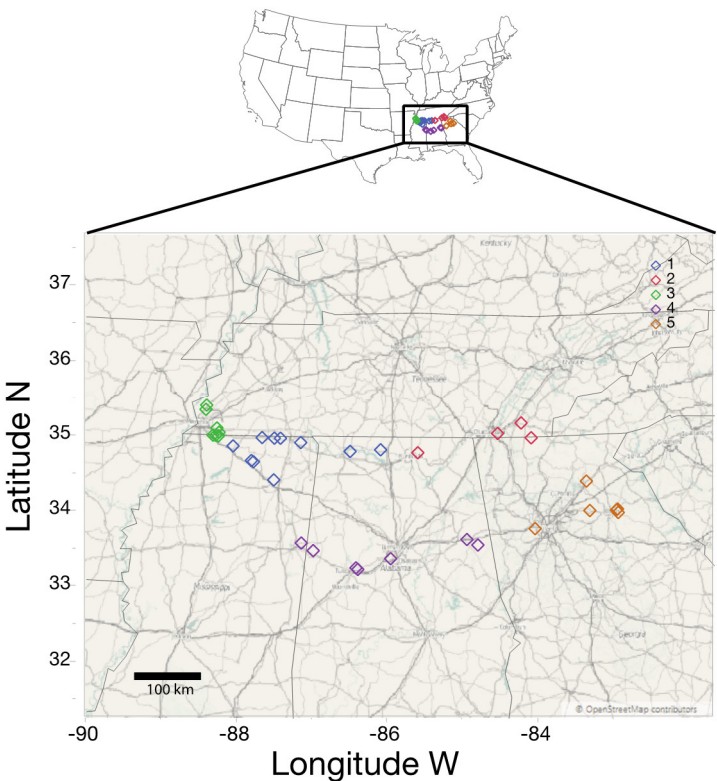

**Figure 1.** Map of sampling sites across the historic range of kudzu infestations in the Southeastern United States. Sampling sites are color coded based on k-means clustering into five broad location clusters used for analysis.

*Sample processing*—To remove fungal epiphytes, spores, hyphal fragments and naked DNA from the surface of the leaf, we agitated leaves for 1 min in sterile in 1% Triton-X 100 (*v:v*) in distilled water [72,73]. This was followed by three washes for 30 s in sterile water. Nine leaf disks were excised using a sterilized handheld paper punch and placed directly into DNA extraction tubes (IBI Soil Extraction Kits, IBI Scientific, Dubuque, IA, USA) (*following* [73]), and stored at −20 °C until processed further.

*Plant, soil, and geographic measurements*—For each leaf, we measured leaf chlorophyll and specific leaf area, and, for each plant, we measured sap nitrite and potassium, as well as soil pH. We also recorded local traffic intensity and GPS coordinates, and we demarcated each of the 38 sites to location clusters (see below). Chlorophyll was measured using a handheld chlorophyll meter (atLEAF CHL chlorophyll meter; atLeaf, Wilmington, DE, USA) and obtained values were converted to SPAD units (following [74]), and total chlorophyll content (mg/cm$^2$) was calculated. Total specific leaf area was quantified following high-resolution photography in an enclosed portable light box with the software

Image J (v.1.15; [75]). Total nitrate ($NO_3^-$) and potassium ($K^+$) concentrations (ppm) were measured per plant from leaf and stem sap following grinding and sap extraction from fresh tissue, using portable LAQUAtwin Meters (LAQUAtwin K-11 and LAQUAtwin $NO_3^-$, Horiba Scientific, Kyoto, Japan). To measure soil pH, we collected soil (4 cm deep) using a hand corer directly underneath collected leaves, and pH was measured with a 1:1 soil: distilled water ratio with a pH electrode (Orion Star A; Thermo Fisher Scientific, Waltham, MA, USA); for a few locations, soils were unable to be collected due to the presence of a shallow hardpan (see Table S1). As kudzu has an affinity for disturbed land, such as roadsides, we measured traffic intensity. For each sampling location, we determined the average annual daily traffic load (AADT—average number of vehicles/road/day) using publicly available data from the Departments of Transportation of each of the respective states. Traffic load is a general proxy for vehicular pollution (including Cd and PAHs; [76]), and vehicle pollution is suspected to be a driver of microbial communities [77]. We selected the busiest road within 100 m of sampling locations and categorized AADT values into 5 categories based on quartiles of the data (Zero [samples not near a road]; Low [Residential roads—5900 daily vehicles]; Moderate [5901–14,075 AADT values]; High [14,076–29,000 AADT values]; and Extreme [greater than 29,001 AADT values]). Further, as spatial variability and geographic location play a large role in structuring fungal endophytic communities [78], we wanted to demarcate our samples into location regions independent of political or arbitrary boundaries, to examine if endophytic communities differ across our sampling sites. To do so, we conducted hierarchical K-means clustering using latitude and longitude values for each sampling location, where minimum membership for each location cluster was set as four (JMP Pro v 14); this identified five location clusters (regions; Figure 1; Table S1). Additionally, after downstream fungal sequencing (see below), OTU2 (97% similarity) was determined to be kudzu in origin rather than fungi. Consequently, we harvested all sequences belonging to OTU2, truncated sequences to 250 bp, and pairwise aligned these sequences, and the resultant distance matrix was used to cluster them into operational taxonomic units (similar to below) at a 99% similarly threshold, to delineate probable plant genotypes (similar to [79]), which resulted in the demarcation of six kudzu genotypes (Table S1). It has been previously demonstrated that kudzu lacks spatial genetic or population structure in its invaded range [38]; therefore, we did not expand on these genotypes for additional analyses. These plant and environmental variables were used in downstream analyses of fungal community data.

*DNA extractions, Amplicon Library Generation, and Sequencing*—Genomic DNA (gDNA) extraction was conducted following IBI kit protocols (IBI soil DNA extraction kit, IBI Scientific, Dubuque, IA, USA) with an additional homogenization step with a Fisherbrand™ Bead Mill 24 Homogenizer (Thermo Fisher Scientific, Waltham, MA, USA) for 1 min at maximum speed. Extracted DNA was quantified using a Nanophotometer N60 (Implen, München, Germany) and DNA was diluted to a working concentration of 10 ng $\mu L^{-1}$. PCRs were conducted using a two-step amplification procedure (following [73]) whereby the fungal Internal Transcribed Spacer region 2 (ITS2) was amplified with the primers nexF-N[3]-fITS7 and nexR-N[3]-ITS4, where fITS7 [80] and ITS4 [81] are fungal ITS2 primers, N[3] represents three ambiguous nucleotides to increase nucleotide diversity, and nexF and nexR are the Nextera sequencing primers. Primary PCRs were conducted in 25 $\mu L$ reactions with the following concentrations: 5 $\mu L$ DNA template (50 ng), 12.5 $\mu L$ 2X Phusion master mix (Thermofisher Scientific, Waltham, MA, USA), 0.5 $\mu M$ (2.5 $\mu L$) of each forward and reverse primer, and 2.5 $\mu L$ molecular grade $H_2O$. PCR parameters were 98 °C for 30 s, 25 cycles of 98 °C for 20 s, 51 °C annealing temperature for 30 s, and 72 °C for 40 s, followed by a final extension at 72° for 10 min; all ramp rates were 1 °C/s (SimpliAmp Thermal Cycler, Applied Biosystems, Foster City, CA, USA). This resulted in a 1° PCR construct of nexF-N[3]-fITS7-{ITS2}-ITS4-N[3]-nexR. After primary PCR generation and amplification verification using gel electrophoresis, secondary PCR reactions (20 $\mu L$) consisted of the forward primer (P5-i5-overlap) and the reverse primer (P7-i7-overlap), where P5 and P7 are Illumina Adaptor sequences, i5 and i7 are 8 bp unique Molecular Identifiers (MIDs), and overlap is the partial nexF or nexR, which is the annealing site for 2° PCRs. The forward- and reverse-barcoded 2° primers were mixed to generate unique dual-barcoded primers (see Table S2) of 10 $\mu M$ (5 $\mu M$ for each primer).



The 2° PCR reactions were 2 μL of 1° PCR product, 10 μL 2X Phusion Master Mix, 0.5 μM of mixed primers, and 6 μL molecular grade $H_2O$. PCR parameters were 98 °C for 30 s, 8 cycles of 98 °C for 20 s, 50 °C for 30 s and 72 °C for 40 s, followed by a final extension at 72° for 10 min. This generated final constructs of P5-i5-nexF-N[3]-fITS7-{ITS2}-ITS4-N[3]-nexR-i7-P7 32 total cycles. Secondary PCR products were cleaned using Axygen AxyPrep Mag PCR breads (Axygen Biosciences, Union City, CA, USA) following a kit protocol modified to use a 1:1 bead-solution-to-reaction-volume ratio [82]. Negative controls (molecular grade water) were included throughout and were free of observable amplification. Final amplicons were quantified using Qubit 3.0 fluorometric assays (dsDNA HS Assay Kit; Thermofisher Scientific, Waltham, MA, USA). Fungal PCR products were pooled at equal concentrations (115 ng per sample). Amplicons were sequenced on one half of a reaction of Illumina MiSeq (300PE) at the Kansas State University's Integrated Genomics Facility (Manhattan, KS, USA). Demultiplexing of the raw sequence provided individual paired fastq files. Sequence data are deposited at the Sequence Read Archive (SRA) at NCBI under the accessions: BioProject (PRJNA626645) and BioSamples (SAMN14646816-SAMN14646929).

*Bioinformatics*—Sequences were processed using the program mothur (v.1.41.1; [83]) generally following [84] with modifications. Briefly, paired sequences were contiged and screened to remove ambiguous base pairs and homopolymers longer than 12 bp and merged into a single fasta file. Sequences were pre-clustered to minimize sequencing induced errors (following [85] as implemented in mothur), and the remaining sequences were screened for chimeric properties (mothur-implemented VSEARCH; [86]) and putative chimera culled. Using a Naive Bayesian classifier [87] against the UNITE non-redundant Species Hypothesis database (v6; [88]), sequences were screened for off-target amplification. OTU2 was defined as kudzu and was used for genotype demarcation. Remaining sequences were demarcated into Operational Taxonomic Units (OTUs) using the mothur-implemented VSEARCH (abundance-based clustering; [86]) with a 97% similarity threshold. OTUs with 10 or fewer sequences globally were considered potentially suspect and culled [89,90]. OTUs were assigned the level of Species Hypothesis where possible (UNITE). While most OTU taxonomic identities are likely correct, relying on autoclassification for taxonomic identities should be cautiously interpreted. After all sequence quality control, we retained $\sim 2 \times 10^6$ high-quality and verified sequence reads.

*Statistical analysis*—Relative OTU richness ($S_{obs}$), diversity (complement of Simpson's diversity index; 1-*D*), and evenness (Simpson's evenness, $E_D$) of fungal endophytes were estimated using an iterative subsampling approach (500 sequences for 1000 iterations), and the mean values were used for all analyses. These diversity estimates were tested against all collected physio-biochemical and geographic measurements using a combination of regression analyses and analysis of variance (ANOVA) models. Diversity and evenness were logit-transformed to meet ANOVA assumptions of normality, whereas richness values were Box–Cox transformed ($\lambda = 2$) prior to analyses. Further, to visualize hierarchical community structure, we used the package *Metacoder* in R (v.0.3.3; [91]). This quantitatively displays numeric data associated with taxa abundance, using the color and size of nodes and edges in a taxonomic heat tree. To examine individual OTU responses to physio-biochemical and geographical factors, we constrained our analyses to the 100 most abundant OTUs, which represent more than 95% of the entire endophytic community. Relative abundances of individual OTUs were analyzed using either linear regression or analysis of variance (ANOVA) models against physio-biochemical and geographic variables where appropriate (using JMP Pro v14). All OTU relative abundance data were tested for normality, and where non-normal, individual Box–Cox transformations were conducted prior to analysis.

To test if plant endogenous attributes (Chlorophyll, $NO_3^-$, $K^+$, leaf surface area, kudzu genotype) and exogenous attributes (soil pH, location, and traffic) play major roles in structuring fungal endophytic communities, we used a permutational multivariate analysis of variance (PerMANOVA; [92]) approach using R (function adonis, package *vegan*; [93]) with 999 permutations. PerMANOVA was conducted on iteratively subsampled (as above) Bray–Curtis dissimilarity values. While PerMANOVA is generally insensitive to concerns with multicollinearity [94], we examined bivariate correlations (Kendall tau)

between explanatory variables and all of these comparisons have correlation coefficients <|0.35|, so these data are not considered multicollinear. To test if endophytic communities are spatially structured and exhibit isolation by distance, we conducted a Mantel (as implemented in mothur) test using Bray–Curtis dissimilarity values and geographic pairwise distances (km) between sampling locations using Kendall rank correlations with 1000 iterations.

Fungal OTUs annotated to species hypothesis or genus levels (UNITE) were classified into functional groups using the FUNGuild database (v 1.1; [95]). Using FUNGuild, primary literature and expert knowledge, we classified the 100 most abundant OTUs, where possible, into either true endophytes or putative pathogens. As putative pathogenic taxa may be pathogens on one plant but endophytes on another, we aimed to investigate the relationships between true endophytes (non-pathogens) and putative pathogenic agents (that have been documented to be pathogenic on Fabaceae, thereby likely pathogenic on kudzu). To do this, we used the nonparametric Kendall's tau rank correlations to measure the associations between all putative pathogenic and endophytic OTUs based on their relative abundances. All statistical analyses were conducted using a combination of JMP, R, and mothur.

## 3. Results

After quality control, denoising, and chimera removal, 595 fungal OTUs were retained and were dominated by members best identified with the phylum Ascomycota (359 OTUs [60.34%] and 86.47% of the total sequences); of these, OTUs identified the class Dothideomycetes (219 OTUs [36.81%] and 76.07% of the total sequences), the order Tremellales (147 OTUs [24.70%] but only 8.55% of the total sequences), and the family Mycosphaerellaceae (54 OTUs [9.07%] and 15.73% of the total sequences), were the most abundant (Figure 2, Table S3, Table S4). Analyses of the effects on diversity estimates by physio-biochemical and geographic attributes indicate uneven drivers of fungal endophytic diversity (Table 1) with nitrate showing a negative correlation with 1-$D$ and $E_D$ but not richness, soil pH showing a positive correlation with only richness and most diversity estimates showing difference with traffic intensity and location evenness. Interestingly, kudzu genotype does not appear to affect diversity estimators.

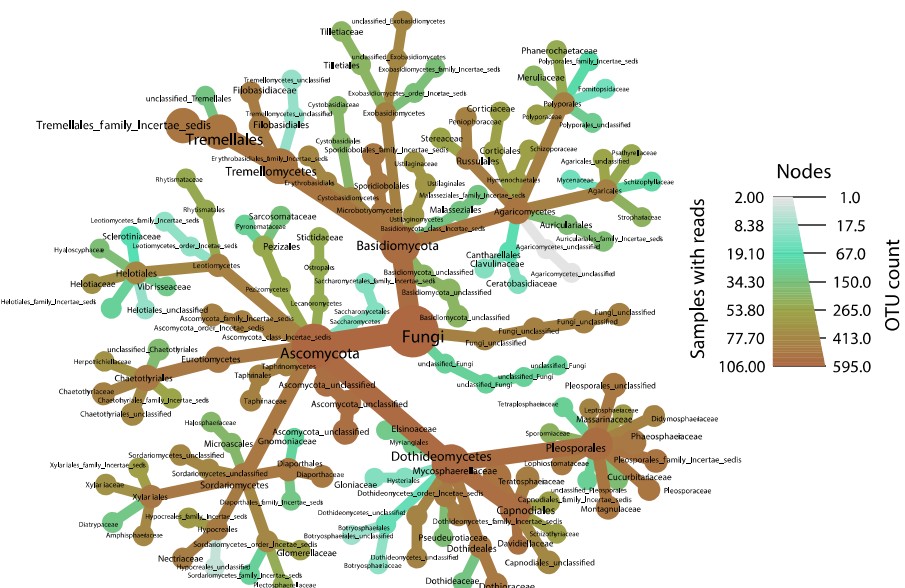

**Figure 2.** Taxonomic distribution of fungal endophytes within sampled kudzu leaves, as visualized using the program Metacoder, resolved to the family level. Nodes are sized proportional to OTU abundances for each taxonomic rank, and OTUs that are more common throughout our samples fall toward the brown end of the color gradient.

**Table 1.** The effects of physio-biochemical and geographical parameters (chlorophyll, leaf area, leaf nitrate, potassium, traffic intensity, location and genotype) on fungal community diversity metrics (diversity, richness, evenness) using regression or ANOVA analyses were applicable. NS indicates a non-significant relationship.

| Tested Variable. | Diversity (1-D) | Richness ($S_{obs}$) | Evenness ($E_D$) |
|---|---|---|---|
| Chlorophyll (mg/cm$^2$) | NS | NS | NS |
| Genotype | NS | NS | NS |
| Leaf area (cm$^2$) | NS | NS | NS |
| *Location* | $F_{4,88} = 12.884, p < 0.001$ | $F_{4,88} = 7.280, p < 0.001$ | $F_{4,88} = 10.423, p < 0.001$ |
| *Nitrate (ppm)* | $t = -3.27, p = 0.0015$ | NS | $t = -2.21, p = 0.0292$ |
| Potassium (ppm) | NS | NS | NS |
| *Soil pH* | NS | $t = 2.50, p = 0.0143$ | NS |
| *Traffic Intensity* | $F_{4,89} = 4.059, p = 0.0045$ | $F_{4,89} = 3.530, p = 0.010$ | NS |

Individual OTU responses (100 most abundant OTUs) to physio-biochemical and geographical measurements indicate that traffic intensity, location cluster, genotype, and soil pH had the strongest impacts on OTU relative abundances (Table 2, Table S5), with these variables significantly impacting 37%, 29%, 12%, and 10% of tested OTUs, respectively. Our data suggest that OTUs within the families Montagnulaceae and Davidiellaceae are the most responsive OTUs to our measured variables (Table S5).

**Table 2.** Percentage of the 100 most abundant OTUs that were significantly responsive to physio-biochemical or geographic measurements (Regression or ANOVA). Where analyses are regression based, the percentage of positive and negative relationships are given parenthetically. Results are presented ranked by the number of OTUs responsive to measurements. Full test statistics are presented in Table S5.

| Environmental Factor | Percentage of Significant OTUs (Positive %, Negative %) |
|---|---|
| Traffic Intensity | 37% |
| Location Cluster | 29% |
| Kudzu Genotype | 12% |
| Soil pH | 10% (8%, 2%) |
| Leaf Chlorophyll | 10% (10%, 0%) |
| Leaf Area | 9% (4%, 5%) |
| Stem and Leaf K$^+$ | 5% (0%, 5%) |
| Stem and Leaf NO3$^-$ | 2% (2%, 0%) |

Foliar endophytic communities of kudzu are structured by multiple factors based on PerMANOVA analyses (Table 3). Our data revealed that the strongest drivers (based on $R^2$ values) are location ($R^2 = 0.152$), plant genotype ($R^2 = 0.129$), and traffic intensity ($R^2 = 0.126$). Interestingly, all tested parameters showed a significant effect on fungal communities, but additional factors each explained less than 5% of community variation (Table 3) with residual analysis suggesting that over 44% of the community variation was unaccounted for in our model. To investigate if kudzu endophytes showed isolation by distance patterns (which we expected, given that location cluster was the factor that accounted for most of the community variation within our model) we used a Mantel test, and this demonstrated that the more distant samples were, the more dissimilar communities become ($\tau = 0.114$, $p < 0.001$).

**Table 3.** Drivers of endophytic communities of Kudzu based on PerMANOVA analysis based on the Bray-Curtis dissimilarity values. Factors are listed in the order of descending $R^2$ values.

| Model Factor | Df | Pseudo-F | $R^2$ | *p* Value |
|---|---|---|---|---|
| Location | 4 | 5.729 | 0.15299 | 0.001 |
| Genotype | 5 | 3.8823 | 0.12959 | 0.001 |
| Traffic Intensity | 4 | 4.7323 | 0.12637 | 0.001 |
| Soil pH | 1 | 6.1513 | 0.04107 | 0.001 |
| Leaf $NO_3^-$ | 1 | 4.7655 | 0.03182 | 0.001 |
| Chlorophyll | 1 | 4.4625 | 0.02979 | 0.001 |
| Leaf Area | 1 | 4.0106 | 0.02678 | 0.001 |
| Leaf $K^+$ | 1 | 2.7078 | 0.01808 | 0.003 |
| Residuals | 95 | | 0.44351 | |
| Total | 113 | | 1 | |

We examined the correlational relationship between putative pathogenic fungi and non-pathogenic endophytes (Figure 3; Table S6). We found numerous endophytes that were positively and significantly correlated with known legume pathogens. It is evident that these co-occur readily. Further, there was not a single negative association between these comparisons (Table S6). We see (Figure 3) that endophytic OTUs that are unresolved within the order Tremellales (Tremellales incertae sedis) and within the family Montagnulaceae make up most of these strong positive associations with putative pathogens. Further, pathogens within the genera *Phoma*, *Alternaria*, *Periconia*, and *Diaporthe* are particularly associated with endophyte occurrence, and, based on the strengths of the correlation coefficients, endophytes within Montagnulaceae and Davidiellaceae mostly strongly and positively co-occur with pathogens (Figure 3). It is important to note that we have no evidence that these putative pathogens (as determined using FUNGuild and primary literature) are actively pathogenic on kudzu, as we selected asymptomatic leaves and cryptic fungal infections are common [96], but it is very likely, as latent foliar infection potential is high in plants, but often depends on pathogen mode of action [97]. Further, there is potential that our identified endophytes (non-pathogenic) may actually be pathogenic agents, as knowledge about disease incidence and activity on kudzu is lacking and several closely related taxa to our endophytes are known to be pathogens in other systems; however, with our data we have no reason to doubt the veracity of functional identifications.

| Putative Endophytes | Putative Kudzu Pathogens | | | | | |
|---|---|---|---|---|---|---|
| | Phoma sp. (OTU3) | Alternaria sp. (OTU 15) | Periconia sp. (OTU25) | Colletotrichum sp. (OTU 75) | Diaporthe sp. (OTU84) | Mycosphaerella sp. (OTU 90) |
| Tremellales Incertae Sedis (9) | 77.8% | 100.0% | 88.9% | 33.3% | 100.0% | 22.2% |
| Montagnulaceae (3) | 66.7% | 66.7% | 66.7% | 33.3% | 66.7% | 33.3% |
| Mycosphaerellaceae (3) | 66.7% | 66.7% | 100.0% | 0.0% | 100.0% | 0.0% |
| Pleosporaceae (2) | 100.0% | 100.0% | 50.0% | 0.0% | 100.0% | 0.0% |
| Ascomycota sp. (1) | 100.0% | 100.0% | 100.0% | 100.0% | 100.0% | 100.0% |
| Chaetothyriaceae (1) | 0.0% | 0.0% | 0.0% | 0.0% | 0.0% | 0.0% |
| Chaetothyriales sp. (1) | 0.0% | 0.0% | 0.0% | 0.0% | 0.0% | 0.0% |
| Davidiellaceae (1) | 100.0% | 100.0% | 100.0% | 100.0% | 100.0% | 100.0% |
| Diaporthaceae (1) | 100.0% | 100.0% | 100.0% | 100.0% | 100.0% | 0.0% |
| Dothideomycetes Incertae Sedis (1) | 100.0% | 100.0% | 100.0% | 0.0% | 100.0% | 100.0% |
| Dothioraceae (1) | 0.0% | 0.0% | 0.0% | 0.0% | 100.0% | 0.0% |
| Filobasidiaceae (1) | 100.0% | 100.0% | 100.0% | 0.0% | 100.0% | 100.0% |
| Phaeosphaeriaceae (1) | 100.0% | 100.0% | 100.0% | 100.0% | 100.0% | 100.0% |
| Taphrinaceae (1) | 100.0% | 100.0% | 100.0% | 100.0% | 100.0% | 100.0% |
| Thioraceae (1) | 100.0% | 100.0% | 100.0% | 100.0% | 100.0% | 100.0% |
| Average Kendall Tau | >0.6 | 0.5-0.6 | 0.4-0.5 | 0.3-0.4 | <0.3 | |

**Figure 3.** Heat map indicating the percentage of significant endophytic OTUs within each family (rows; total number of tested OTUs with families are shown parenthetically) that are significant and positively associated with pathogens (columns, with OTU number presented parenthetically) Average Kendall's tau correlation coefficients for all significant correlations are used for color demarcation (see last row).

## 4. Discussion

In this novel study of kudzu, we investigated the community composition of foliar fungal endophytes and examined the potential contribution of biophysiochemical and geographical factors on this community assembly across 38 different sites in the Southeast United States. Our findings revealed that endophytic fungal communities of kudzu were mainly composed of Ascomycotan species (Table S3), which is consistent with other studies investigating community compositions of phyllospheric fungi [98,99]. Within Ascomycota, the majority of taxonomically resolved OTUs belonged to the Dothideomycetes (Figure 2). Dothideomycetes, the most speciose class of fungi, are commonly associated with plants and are often beneficial to ecosystem health and carbon cycling. Saprobic Dothideomycetes generally decompose cellulose or other plant-derived carbohydrates [100,101]. Tremellales (phylum Basidiomycota) were the second-most abundant class of fungi observed in our samples. Tremellales are interesting here, as they often include both teleomorphic and anamorphic species and/or growth forms. Teleomorphic Tremellales are mainly mycoparasites with macro-fruiting bodies, while anamorphic Tremellales often have yeast-like growth forms [102]. This is particularly interesting, as it suggests a capacity for micro-growth forms and fungal–fungal interactions within kudzu leaves. *Tremella*, an abundant yet polyphyletic genus within the Tremellaceae [103], includes mainly mycoparasitic species [104]. This further indicates the strong potential of fungi–fungi interactions playing major roles in the presence of pathogenic taxa occurrence. It may be that these *Tremella* spp. parasitize pathogen inhibitors, which may lead to relaxation of direct pathogen antagonisms, but this has not been explicitly tested here. Additional work investigating mechanistic interactions between *Tremella* and other taxa are needed.

Based on regression analyses, we found a positive relationship between fungal diversity metrics with soil pH and traffic intensity (Table 1). Kudzu plants in heavily trafficked locations show higher diversity and richness. One explanation is that vehicular exhaust is a major ambient source of Polycyclic Aromatic Hydrocarbon (PAHs) in urban areas [105], and growing evidence indicates that some microbial endophytes are able to biodegrade the PAHs; therefore, endophyte assemblages may be affected by exposure to air pollution, which can act as a niche differentiation vector which facilitates this increased diversity. It has been shown that many fungi have enzymes that can degrade or utilize PAHs [106]. It has been demonstrated that vehicle exhaust emissions influence leaf bacterial communities [107]; however, data regarding vehicle exhaust influence on leaf endophytic fungi are sparse and should be confirmed. We found that soil pH was significantly and positively correlated with richness. The pH of the soil has already been reported to impact the soil- and plant-associated microbial community diversity [108]. One mechanism by which the soil pH can affect soil microbial populations is through modifying nutrients availability in the soil which ultimately impact leaves' nutrients and metabolite contents [109,110]. Nutrient availability in the soil is one of the factors that influences foliar nutrient concentration, which directly or indirectly accounts for some leaf traits, such as cell extension, as well as membrane function and stability. González-Teuber et al. [111] reported that leaf resistance traits such as cell-wall polysaccharides, leaf toughness, flavonoids, anthocyanins, terpenoids and chitinases are impacted by local nutrient conditions, which likely has a major impact on leaf endophyte colonization and community composition.

While we found strong impacts of soil pH and traffic intensity on some of the fungal community diversity metrics, we also aimed to quantify the impacts of physio-biochemical and geographical factors on individual OTU responses. To do this, we conducted a series of regression analyses on the 100 most abundant OTUs, representing 95% of the entire community by relative abundance (Table 2; Table S5). Our results show that 37% and 29% of the OTUs were responsive to traffic intensity and location cluster, respectively. Among the OTUs responsive to traffic intensity and location cluster, OTUs within the families Davidiellaceae (particularly *Davidiella* spp.—polyphyletic with some *Cladosporium*) and Tremellales incertae sedis (particularly *Cryptococcus* spp.) were often responsive to one or both variables.

Consistent with a large body of literature (reviewed in [19,112,113]), we found that all measured parameters, such as soil pH, leaf nitrate, chlorophyll, specific leaf area, and leaf potassium, significantly shaped the total fungal endophyte community (Table 3); however, location, genotype and traffic intensity were among the strongest drivers of the kudzu endophytic communities. Li et al. suggested that plant genotype shapes plant phenotype, which can lead to host specificity to microbes [114] due to plant genotypic variation in the release of different quantities of organic compounds as root exudates, which affects the process of fungal recruitment into the rhizosphere [115,116]. The impacts of location on foliar fungal communities might be the result of significant dependency of the biogeographic distribution of endophytic microbes on physical and chemical properties of soil [116,117], environment conditions (e.g., climate and topography), and even land cover.

It is worth a brief discussion of the impacts of soil pH on endophytic community assembly, as it may seem that these should be unrelated. While we know relatively little about the mechanisms that connect below-ground chemistry to above-ground endophytes, it has been known for decades that experimental changes in soil acidity impact fungal leaf endophytes [118]. The most accepted reason for this relationship has to do with foliar endophytic modulation of soil processes and stoichiometry, particularly as related to phosphorous modulation [119] and mycorrhizal colonization [120], both of which can impact pH. This potential rhizosphere modulation is not yet fully resolved [121] and additional research in needed to disentangle directionality of effect.

In our exploration of the potential interactions between fungal endophytes and putative pathogenic agents, we found that there were many significant positive associations between specific OTUs (Figure 3), and there were no significant negative associations. Among the genera that are often pathogenic on leguminous plants and ones we identified on kudzu, *Alternaria*, *Diaporthe*, *Cercospora* and *Colletotrichum* are known as causative agents of severe diseases on some of the crops, such as Cercospora leaf blight on soybean (*Cercospora kikuchii*) [122], Alternaria leaf spot on fava beans (*Alternaria alternata*) [123] and Anthracnose on beans (*Colletotrichum lindemuthianum*) [124]. According to Sun et al., 2005, *Colletotrichum* sp. and *Cercospora* sp. can cause leaf spot diseases on Kudzu [37]. OTUs within the families Montagnulaceae and Tremellales incertae sedis had the most significant positive correlations with putative pathogenic fungi (Figure 2). These results are important and novel in two main ways: (1) there is a dearth of investigations of pathogenic agents on kudzu, which is unfortunate, as kudzu has been demonstrated to be a major pathogen spillover vector [25,27] onto crops, which has been known since 1930 when Higgins [125] demonstrated halo blight of green beans on kudzu; this work greatly expands the current understanding of which pathogens found on kudzu might pose major threats to leguminous crops, and (2) this hints at a complex of endophytic–pathogen interactions that likely play major roles in pathogenicity on kudzu, and if these can be confirmed experimentally, will provide endophyte–pathogen co-inoculation targets for the development of an integrated way to suppress kudzu populations, which, with concurrent herbicide usage, might drastically increase the efficacy of current management strategies. Additional work is needed to confirm these interactions.

This highlights the potential role for endophytes in kudzu disease facilitation and the potential that exists for harnessing these endophytes for the benefit of sustainable agriculture. Many endophytes have been demonstrated to have different ecological interactions, including facilitation, antagonism, or neutral impacts on different plant pathogens [67,68]. Facilitation occurs when one microorganism enhances the development or growth of another. This facilitation may be due to the theory that different species may have similar host requirements and a lack of competitive exclusion [126]. Interactions between endophytes and pathogens that facilitate disease development and pathogenicity have not been extensively studied. Pathogens might exploit endophytes to enhance their pathogenicity, and this connection might be driven by production of a plethora of secondary metabolites by endophytes [67] which may directly or indirectly (via inhibition of a mycoparasite, for instance) benefit the pathogen. Pathogen–endophyte associations have been demonstrated to increase the severity of diseases in some invasive plants [127] but systematic investigation of associated mechanisms is

currently lacking. Further, endophytes associated with pathogens may similarly evade or overcome fungal-resistance mechanisms [128] and can contribute to the differentiation and/or sporulation of the fungal pathogen [129,130].

Until recently, the potential contribution of endophytes in kudzu disease development and the consequences of fungal endophytes and pathogenic species co-occurrences on kudzu has been largely overlooked. We believe this is an area of research that is a promising tool for the development of integrated management strategies of invasive plants, and additional research into these patterns and associated mechanisms is sorely needed. This study also suggests that there may be additional opportunities to investigate bacterial endophytes of kudzu with the aim of invasive control. A remaining challenge is to identify core endophytes that modulate pathogenic fungi to facilitate disease development in invasive plants, which may guide us to microbial-based invasive plant management.

## 5. Conclusions

We found fungal community composition within kudzu leaves was significantly structured by location, host genotype, traffic intensity, and less by soil pH, leaf nitrate, and potassium and leaf area. Furthermore, a significant correlation was detected among community diversity matrices with traffic intensity and soil pH. Future work should use experimental manipulations to disentangle other co-varying predictors, such as leaf secondary metabolites content. Given the impact of kudzu, including increased ozone pollution, as a reservoir of pathogens for regionally important crops including soybean, alterations in soil biogeochemical cycling, and out-competing native plants, this study provides a platform for understanding how kudzu pathogen–endophytic communities are inextricably linked with environmental and geographical factors at the ecosystem level toward an aim of mycobiome manipulations to facilitate pathogens to suppress kudzu growth.

**Supplementary Materials:** The following are available online at http://www.mdpi.com/1424-2818/12/5/185/s1, Table S1: Sampling locations with associated physiochemical data, Table S2: Primer and MID sequences, Table S3: Taxonomic distribution of OTUs across taxonomic ranks, Table S4: Full OTU information and with functional guilds, taxonomic identifications and OTUxSample matrix. Table S5: Results of analysis of OTUs against collected physiochemical and geographic data, Table S6: Results of endophyte–pathogen correlational analysis.

**Author Contributions:** Conceptualization, design, and implementation was conducted by M.S. and S.P.B., statistical analyses were conducted by M.S., and M.S. and S.P.B. wrote and edited the manuscript. All authors have read and agreed to the published version of the manuscript.

**Funding:** This research was funded in part by the Agriculture and Food Technologies Cluster of the FedEx Institute of Technology, The University of Memphis. Additional funding was provided by the Department of Biological Science at The University of Memphis.

**Acknowledgments:** The authors wish to thank Mark Weaver for discussions about kudzu pathogens and Avery Tucker for his assistance with data visualization. This work is part of the doctoral dissertation of M.S. We express thanks to the three reviewers whose comments greatly improved this manuscript.

**Conflicts of Interest:** The authors declare no conflict of interest.

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
