# Peer review of "Drivers of Foliar Fungal Endophytic Communities of Kudzu (Pueraria montana var. lobata) in the Southeast United States"

_diversity, doi:10.3390/d12050185_

Round 1
Reviewer 1 Report
The authors investigated the fungal endophyte communities in kudzu, an invasive plant species in the US, and the characteristics of host plants and environments affecting them. Using high-throughput sequencing of ITS2 amplicons, they revealed the diversity and composition of fungal endophyte communities in the 3 x 38 leaf samples collected from different regions. The OTU diversity, the distribution of abundant fungal OTUs, and the OTU composition were mainly determined by the traffic intensity, localities, and genotypes of host plants. Some pathogenic fungi that are known to cause disease in crops likely interact with some specific fungal endophytes in kudzu, and the interactions could be useful to control the growth of kudzu populations. The study provided original insights into the fungal endophytic communities in kudzu leaves and can contribute to build management strategies of kudzu in the US. I have only two concerns about the current manuscripts. (1) Although the authors showed the significant effect of host genotype on the fungal communities, the diversity and spatial distribution of the host genotypes themselves is not clear. How many genotypes (clones) were found in total? If there is a population structure in the host genotypes, that should be accounted for in the statistical analyses. (2) In the PerManova, please examine and show in the MS if multicollinearity exists among the explanatory variables, which would bias the analysis.Author Response
We appreciate the through and supportive review. I provide point by point responses below the original reviewer statements as indicated by ">>"
The authors investigated the fungal endophyte communities in kudzu, an invasive plant species in the US, and the characteristics of host plants and environments affecting them. Using high-throughput sequencing of ITS2 amplicons, they revealed the diversity and composition of fungal endophyte communities in the 3 x 38 leaf samples collected from different regions. The OTU diversity, the distribution of abundant fungal OTUs, and the OTU composition were mainly determined by the traffic intensity, localities, and genotypes of host plants. Some pathogenic fungi that are known to cause disease in crops likely interact with some specific fungal endophytes in kudzu, and the interactions could be useful to control the growth of kudzu populations. The study provided original insights into the fungal endophytic communities in kudzu leaves and can contribute to build management strategies of kudzu in the US. I have only two concerns about the current manuscripts.
(1) Although the authors showed the significant effect of host genotype on the fungal communities, the diversity and spatial distribution of the host genotypes themselves is not clear. How many genotypes (clones) were found in total? If there is a population structure in the host genotypes, that should be accounted for in the statistical analyses.
>>This is a great point. We added language to clearly indicate how may genotypes were demarcated. Further, it has previously been demonstrated that kudzu have very little geographic patterning and population structure in its invaded range in the US (Bentley & Maurcio, 2016, 103: 1499-1507). Therefore, we concluded that population structure (which is essentially random across the US) does not need to be accounted for in our statistical analyses.
(2) In the PerManova, please examine and show in the MS if multicollinearity exists among the explanatory variables, which would bias the analysis.
>>PerMANOVA is insensitive to variable correlation structure (Anderson & Walsh, 2013, Ecological Monographs, 83: 557-5774; PERMANOVA + for Primer software manual (we did not do PerMANOVA using this program, but it was written my Marti Anderson, who developed PerMANOVA)). So, we disagree with the assertion that it will bias the analyses. However, we agree with the reviewer’s suggestion that the relationship between these explanatory variables should be addressed. We now include language examining correlations between variables to be more transparent. We find that of all the comparisons, the largest correlation coefficient =-0.33 (Kendall tau), much lower than what is generally considered to be an indication of multicollinearity. Thus, we can conclude that multicollinearity is not a concern here.
Reviewer 2 Report
Sharhtash and Brown’s manuscript examining the fugal microbiota on kudzu was a pleasure to read. I think it is a well-conceived experiment and the manuscript was well written. I have only a few recommendations that I would like the authors to consider prior to publication and highlight some minor comments that I hope will improve the manuscript.
Recommendations:
- This is rather minor, but with regard to your goals, I have a few suggestions: First, above on lines 107-108 you define endophytes as taxa that are asymptomatic. However, you are interested in species that are also pathogenic. As such, are you only interested in epiphytes? Second, are you elucidating the major drivers of foliar fungal kudzu epiphytes, or examining drivers that influence epiphytic assemblages on kudzu across …..? Third, in your second goal, you are interested in both facilitation and inhibitory effects between true endophytes and potential pathogens.
- Lines 146-154, need some attention. First, locations, sampling sites and sites, I believe all refer to the same thing. I would choose one word and use it consistently. Line 149 needs to be reworded “ where feasible leaves were sampled”. I guess I am caught up on “feasible”. I would use “accessible”/ Line 150. How were sampling sites really selected to maximize spatial variation? I am left wondering what the prevalence of symptomatic leaves is and how fungal assemblages may differ on those leaves? I know it is not your study question, but a little info may spur future research using a different approach.
- Lines 185-187: I have been racking my brain about this approach. To me it seems circular. First, you use K-means clustering to put sites into “geographical locations”, then later you test if bacterial species differ among locations. It seems that you already know the answer. I am not sure how to better deal with this. Did you have any other a-prior hypotheses about geographical locations that do not use the bacterial data to define these? Could you remove geographical location from your multi-regression and just use your mantel test to discuss geographic locations? Also, I would love a figure that shows how the locations are distributed.
- End of methods: At the end I had many questions that were addressed later, but might be worth including some description of how you might go about interpreting results. For example, at this point, I did not know if a fungal species could be an epiphyte on kudzu, but a pathogen on another plant species. Some description of the complexities and inclusion in the goals (see suggestion 1) would help clarify this for the readers earlier. Everything becomes clear in the discussion, but I was struggling with all of this (what is a endophyte, how many fungi would not be endophytes, are pathogens always pathogens –even on different species, is a pathogen an endophyte when there are no signs of its effect?) before I got to the discussion.
Minor Comments:
- Lines 63-64: I suggest changing the order here: For example B. vulgaris, a widespread invasive barberry….
- Line 66: Class A is not defined. I suggest defining what this is or removing this designation.
- Lines 66-67: Something is going on here. You are either missing a verb in the first part of the compound sentence, or you need to delete the “and”, or I am reading this wrong, but whatever the case I suggest editing this to help the flow.
- Line 279: I am not sure I follow “where unambiguously possible” – what does this mean specifically?
- Line 300: I love this figure, but figures typically do not have titles. Thoughts?
- Line 435: citation?
Hope this helps.
Author Response
We appreciate the through and supportive review. I provide point by point responses below the original reviewer statements as indicated by ">>"
Comments and Suggestions for Authors
Sharhtash and Brown’s manuscript examining the fugal microbiota on kudzu was a pleasure to read. I think it is a well-conceived experiment and the manuscript was well written. I have only a few recommendations that I would like the authors to consider prior to publication and highlight some minor comments that I hope will improve the manuscript.
Recommendations:
- This is rather minor, but with regard to your goals, I have a few suggestions: First, above on lines 107-108 you define endophytes as taxa that are asymptomatic. However, you are interested in species that are also pathogenic. As such, are you only interested in epiphytes? Second, are you elucidating the major drivers of foliar fungal kudzu epiphytes, or examining drivers that influence epiphytic assemblages on kudzu across …..? Third, in your second goal, you are interested in both facilitation and inhibitory effects between true endophytes and potential pathogens.
>>We apologize if out language was not clear – we are defining endophytes following Stone et al 2000, whereby they are any fungi that line within plant tissue (which excludes epiphytes, we specifically wash epiphytes off before processing to limit our analyses to only internal fungi) and are asymptomatic (but there could be latent or opportunistic pathogens that have yet to exhibit symptomology). This is why we only selected visually uninfected leaves for collection. Finally, you are correct, we were examining true endophytes that may be faciliatory or inhibitory of pathogens. Language has been changed throughout to make these points more explicit.
- Lines 146-154, need some attention. First, locations, sampling sites and sites, I believe all refer to the same thing. I would choose one word and use it consistently. Line 149 needs to be reworded “ where feasible leaves were sampled”. I guess I am caught up on “feasible”. I would use “accessible”/ Line 150. How were sampling sites really selected to maximize spatial variation? I am left wondering what the prevalence of symptomatic leaves is and how fungal assemblages may differ on those leaves? I know it is not your study question, but a little info may spur future research using a different approach.
>>We agree and use sampling sites here and throughout. Line 149 was changed to accessible per suggestion. We struggled to address the spatial variation comment to get our thoughts across in a succinct manner. We meant by this that we specifically did not sample some sites we found if they were geographically close to previous samples because we wanted to avoid spatial autocorrelation concerns. However, after some more thought, I determined that this line adds very little and elicits confusion, so I omitted it from the revision. Further, language was included to give the reader a better understanding on pathogen incidence in sampled sites, per suggestion.
- Lines 185-187: I have been racking my brain about this approach. To me it seems circular. First, you use K-means clustering to put sites into “geographical locations”, then later you test if bacterial species differ among locations. It seems that you already know the answer. I am not sure how to better deal with this. Did you have any other a-prior hypotheses about geographical locations that do not use the bacterial data to define these? Could you remove geographical location from your multi-regression and just use your mantel test to discuss geographic locations? Also, I would love a figure that shows how the locations are distributed.
>>We appreciate this comment. We settled on K-means clustering of locations as we intended to examine if fungal endophyte communities differed with location, but we were not satisfied with the examining latitude and longitude changes, as analyzing these separately, misses aspects of spatial ecology. This we identified locations categories using k-means clustering such that identifies were not influenced by political boundaries or any other biases. Further, we feel using the Mantel test really only tests septation by distance, but this is not necessarily examine geographic patterns. We have revised the language to clarify this, we hope this make more sense. Also, per suggestions, we have included a new figure (Figure 1) that show the spatial distribution of sampling sites.
- End of methods: At the end I had many questions that were addressed later, but might be worth including some description of how you might go about interpreting results. For example, at this point, I did not know if a fungal species could be an epiphyte on kudzu, but a pathogen on another plant species. Some description of the complexities and inclusion in the goals (see suggestion 1) would help clarify this for the readers earlier. Everything becomes clear in the discussion, but I was struggling with all of this (what is a endophyte, how many fungi would not be endophytes, are pathogens always pathogens –even on different species, is a pathogen an endophyte when there are no signs of its effect?) before I got to the discussion.
>>We have altered this language per suggestions to better clarify here the idea that some putative pathogenic taxa may not be pathogenic on all plant hosts.
Minor Comments:
- Lines 63-64: I suggest changing the order here: For example B. vulgaris, a widespread invasive barberry….
>>Sentence structure changed as suggested.
- Line 66: Class A is not defined. I suggest defining what this is or removing this designation.
>>Class a was removed here for clarity. Thanks for the suggestion.
- Lines 66-67: Something is going on here. You are either missing a verb in the first part of the compound sentence, or you need to delete the “and”, or I am reading this wrong, but whatever the case I suggest editing this to help the flow.
>>We apologize for our oversight here. We have altered as suggested.
- Line 279: I am not sure I follow “where unambiguously possible” – what does this mean specifically?
>>We have deleted the word ‘unambiguously’ here as that word, funny enough, made this sentence ambiguous. What is meant here is not all fungi do we fully understand functional roles, thus, we may not be able to call a taxon a true endophyte. We think this is much more clear.
- Line 300: I love this figure, but figures typically do not have titles. Thoughts?
>>We have removed this title and altered the legend to reflect that these were resolved to the family level.
- Line 435: citation?
>>Blast, that is embarrassing. Thank you for the catch, this has been corrected.
Hope this helps.
>>It most certainly did! Thanks.
Reviewer 3 Report
The paper is an contributes important information on the endophytes of an invasive plant in the US. Overall, it is well written with a few editorial changes listed below:
Line 53: deletion
Line 57-59: Please rephrase the sentence. It is very cumbersome to read.
Line 65: Should be teleomorphic. Are you sure that is the correct terminology for teliospores.
Line 245: non-fungal
Line 247: VSEARCH
Line 300: within sampled kudzu leaves
Line 311: add comma. OTUs, respectively.
Line 320: PerMANOVA
Line 349: correlation coefficients
Line 356: remove extremely
Line 373: remove and
Line 381: Should be “abundant”
Line 410, add comma. ..cluster, respectively.
Line 451: replace vastly with extensively
Table S5: Check spelling of nitrate.
My biggest concern is that the data is over-interpreted.
Line 383-385: This is a very broad sweeping statement, based on little evidence. I think it is a little premature to state this. For instance, have you identified potential pathogen inhibitors? Would they be potential hosts of Tremella?
Line 387: It is not clear from the paper how the soil pH influence the endophyte communities. I would suggest that this is clarified. The evidence presented seems to be anecdotal and speculative. How large a difference in pH will affect the community, and if what way. I have the same concern about the correlation with pollution levels. It is assumed that there is elevated pollution, but did you verify it?
Other questions that I have:
- How did the variation within sites, compared to between sites?
- The ITS2 region is not reliable for species identification, and should be cautiously interpreted.

Author Response
We appreciate the through and supportive review. I provide point by point responses below the original reviewer statements as indicated by ">>"
The paper is an contributes important information on the endophytes of an invasive plant in the US. Overall, it is well written with a few editorial changes listed below:
Line 53: deletion
>>Thanks for the catch. Has been deleted as recommended.
Line 57-59: Please rephrase the sentence. It is very cumbersome to read.
>>We have rewritten this and reduced verbiage.
Line 65: Should be teleomorphic. Are you sure that is the correct terminology for teliospores.
>>The review is indeed correct on the spelling of teleomorphic, we have restricted this sentence and eliminated this word all together.
Line 245: non-fungal
>>Thanks for this catch. It has been changed.
Line 247: VSEARCH
>>Changes to be correct.
Line 300: within sampled kudzu leaves
>>Corrected.
Line 311: add comma. OTUs, respectively.
>>Corrected. Thanks.
Line 320: PerMANOVA
>>Corrected.
Line 349: correlation coefficients
>>Corrected.
Line 356: remove extremely
>>Fair point, there cannot be levels of a binary statement.
Line 373: remove and
>>Removed as suggested.
Line 381: Should be “abundant”
>>Corrected.
Line 410, add comma. ..cluster, respectively.
>>Corrected
Line 451: replace vastly with extensively
>>Changed as suggested.
Table S5: Check spelling of nitrate.
>>Spelling has been corrected. We appreciate the detailed read through.
My biggest concern is that the data is over-interpreted.
Line 383-385: This is a very broad sweeping statement, based on little evidence. I think it is a little premature to state this. For instance, have you identified potential pathogen inhibitors? Would they be potential hosts of Tremella?
>>We understand. We have not tested these interactions here, our data suggest that these interactions may be occurring, but it is our ongoing work that is testing these relationships. We rephrase this to prevent over-extending our levels of inference. We hope this revised statement remedies this concern. It now reads “It may be that these Tremella spp. parasitize inhibitors pathogen inhibitors, which may lead to relaxation of direct pathogen antagonisms, but this has not been explicitly tested here. Additional work investigating mechanistic interactions between Tremella and other taxa are needed.”
Line 387: It is not clear from the paper how the soil pH influence the endophyte communities. I would suggest that this is clarified. The evidence presented seems to be anecdotal and speculative. How large a difference in pH will affect the community, and if what way. I have the same concern about the correlation with pollution levels. It is assumed that there is elevated pollution, but did you verify it?
>> Generally, it is not well known why pH influences foliar endophytes or if it is the endophytes that modulate soil. We now include a brief discussion point on this as we feel it is an interesting concept that deserves more investigation. “It is worth a brief discussion on the impacts of soil pH on endophytic community assembly as it may seem that these should be unrelated. While we know relatively little about the mechanisms that connect below ground chemistry to above ground endophytes, it has been known for decades that experimental changes in soil acidity impacts fungal leaf endophytes [120]. The most accepted reason for this relationship has to do with foliar endophytic modulation of soil processes and stoichiometry, particularly as related to phosphorous modulation [121] and mycorrhizal colonization [122], both of which can impact pH. This potential rhizosphere modulation is not yet fully resolved [123] and additional research in needed to disentangle directionally of effect.”
Further, it has been better demonstrated pollution levels impact endophyte, often through local toxicity, or as a selection for shifting communities toward those than can utilize PAH (for instance). We try to clarify this throughout.
Other questions that I have:
- How did the variation within sites, compared to between sites?
>>We are uncertain exactly what the reviewer is asking here. Generally speaking, there was not correlations between locations and other measured parameters (as addressed by comments from another review) and exampled on in the MS in discussion about multicollinearity. Because of this, we did not find any major differences between measured variables within and across sites. We actually think this bolsters our argument of measured effects on endophyte communities.
- The ITS2 region is not reliable for species identification, and should be cautiously interpreted.
>>We take issue with the assertion that ITS2 is not reliable for species identification as there are numerous publications that show that ITS2 is as good (and better in some cases) than ITS1 for fungal metabarcoding studies. Further, given how these OTUs were taxonomically identified using a Bayesian classifier against UNITE SH which has full ITS2 sequences, Taxonomic IDs should be same same as would be done using ITS1. However, this brings up a good point. Any metabarcoding study, regardless of targeted loci, may suffer from incorrect IDs. Therefore, ALL identifications should be cautiously interpreted – we specifically add language throughout the MS to indicate that these are a best guess and are not absolute taxon identifications.
Round 2
Reviewer 3 Report
I am satisfied that the authors adequately addressed all concerns. I have no further comments and suggestions.